# Recombinant Spider Silk Fiber with High Dimensional Stability in Water and Its NMR Characterization

**DOI:** 10.3390/molecules27238479

**Published:** 2022-12-02

**Authors:** Tetsuo Asakura, Hironori Matsuda, Akira Naito, Hideyasu Okamura, Yu Suzuki, Yunosuke Abe

**Affiliations:** 1Department of Biotechnology, Tokyo University of Agriculture and Technology, 2-24-16 Nakacho, Koganei, Tokyo 184-8588, Japan; 2Department of Applied Chemistry and Biotechnology, Graduate School of Engineering, University of Fukui, 3-9-1 Bunkyo, Fukui 910-8507, Japan; 3Spiber Inc., 234-1 Mizukami, Kakuganji, Tsuruoka 997-0052, Japan

**Keywords:** recombinant spider silk protein, dimensional stability, ^13^C NMR spectroscopy

## Abstract

Spider dragline silk has unique characteristics of strength and extensibility, including supercontraction. When we use it as a biomaterial or material for textiles, it is important to suppress the effect of water on the fiber by as much as possible in order to maintain dimensional stability. In order to produce spider silk with a highly hydrophobic character, based on the sequence of ADF-3 silk, we produced recombinant silk (RSSP(VLI)) where all QQ sequences were replaced by VL, while single Q was replaced by I. The artificial RSSP(VLI) fiber was prepared using formic acid as the spinning solvent and methanol as the coagulant solvent. The dimensional stability and water absorption experiments of the fiber were performed for eight kinds of silk fiber. RSSP(VLI) fiber showed high dimensional stability, which is suitable for textiles. A remarkable decrease in the motion of the fiber in water was made evident by ^13^C solid-state NMR. This study using ^13^C solid-state NMR is the first trial to put spider silk to practical use and provide information regarding the molecular design of new recombinant spider silk materials with high dimensional stability in water, allowing recombinant spider silk proteins to be used in next-generation biomaterials and materials for textiles.

## 1. Introduction

Spider dragline silks have attracted much attention as a potential source of next-generation biomaterials and textiles because of their outstanding mechanical properties and biocompatibility [1,2,3,4,5,6]. The spider silks have a unique property, supercontraction, which occurs in the hydration process, i.e., interaction with water causes the spider dragline silk fiber to contract by up to 50% in length and to transition from glassy to rubbery phases [7,8,9,10,11]. However, this latter characteristic is not suitable for use in biomaterials, because biomaterials are generally used in water, and therefore, the maintenance of the dimensional stability of biomaterials in water is required [3,4,5]. In addition, for textiles, it is generally required to overcome some inferior characteristics of the fiber, i.e., low water repellency, vulnerability to friction, low dimensional stability, being prone to wrinkles, and so on, and therefore, dimensional stability in water is also necessary [12,13,14].

All spider silks are mainly composed of spidroins. The best characterized silk is the major ampullate silk. It is known that the assembly of major ampullate spidroin 1 (MaSp1) and spidroin 2 (MaSp2) into a fiber demonstrates outstanding mechanical properties [1,2,3,4,5,15,16,17]. The repetitive domains of MaSpl protein are mainly composed of polyalanine (poly-Ala), which forms antiparallel β-sheets (AP-β) in a crystalline region, and Gly-rich regions, which form an amorphous region. The former region has been considered as the origin of high fiber strength, and the latter as the source of the high elasticity of spider dragline silk fiber [1,2,3,4,5]. 

Because of their cannibalistic behavior, we cannot farm spiders. Furthermore, collecting the silk fiber from spiders is time consuming. Therefore, silk genes have been transferred from spiders to other host organisms to construct recombinant spider silk proteins [18]. In our previous work [19,20,21,22], recombinant spider silk protein (RSP) based on the sequence of ADF(*A. diadematus* fibroin)-3 silk from the European garden spider *Araneus diadematus* was produced with *E. coli.* The two major ampullate silk components of *A. diadematus* are historically referred to as ADF-3 and -4 [18]. The primary structure and amino acid composition of RSP are shown in Appendix A [19]. In addition, another spider silk protein named RSSP(QQQ), with a slightly modified primary structure of RSP was also produced using *E. coli* [23]. We used the abbreviation RSSP(QQQ) in reference to the focus on three types of Q. The primary structure and amino acid composition of RSSP(QQQ) are shown in Appendix A [23]. Then, the regenerated fibers of these recombinant spider silk proteins were prepared by the wet spinning process, and the secondary structures of the fibers were investigated using ^13^C solid-state NMR methods.

In this paper, a new recombinant spider silk protein, RSSP(VLI) (Figure 1), based on RSSP(QQQ) [23], where all QQ sequences are replaced by VL while single Q is replaced by I, is produced with *E. coli* to overcome a critical defect: the low dimensional stability of RSSP(QQQ) fiber in water. In addition, to clarify the structural and dynamical change in the fiber formation process of the recombinant silk protein at the molecular level, solid-state NMR was used.

Because the recombinant spider silk proteins are generally obtained in a powder state, a spinning process to prepare the fibers is required in order to use them widely, and the wet spinning method in particular is frequently used [24]. The dissolution of the powders is a critical step in producing the fibers. The spinning solvents used previously for native spider silks [25] and recombinant spider silks [26,27,28,29,30,31,32] were mainly hexafluoroisopropanol (HFIP) and water. More recently, by learning the spinning system of native spiders, water and aqueous buffers were used for preparing recombinant spider silk fibers with both N-terminal and C-terminal domains [33,34,35,36,37,38,39,40,41]. The relatively small molecular weight makes the recombinant spider silk soluble in water, but the strength of the fiber is quite low at present. The RSSP(VLI) powder was insoluble in water, as well as RSSP(QQQ) powder. In addition, there were many voids in the fiber prepared from the HFIP solution of RSSP(QQQ) [23], which was not suitable for textiles.

On the other hand, formic acid (FA) [42,43,44,45,46,47] and CaCl_2_-FA [48,49,50,51] have been used as the spinning solvents for regenerated *B. mori* silk fibroin (RSF), and excellent physical properties have been obtained. Thus, we select FA as the spinning solvent [21,22,23,42,52,53]. FA induces chemical modifications, namely, the formylation of proteins [21,22,23,54,55,56,57]. In addition, the selection of the coagulation solvents is also important. Thus far, alcohols such as methanol (MeOH), ethanol, and isopropanol have been used [24]. We select MeOH as the coagulation solvent [21,22,23,24,25,26,27]. Previous reports have shown the esterification of Ser residue in recombinant spider silk proteins incubated in FA [21,22,23]. Thus, first, the formylation of Ser residue in RSSP(VLI) was studied using the ^13^C solution NMR method. The mechanical properties of the RSSP(VLI) fibers were examined before and after hydration treatments. Then, dimensional stability was determined for the RSSP(VLI) and RSSP(QQQ) fibers to examine changes in the stability when all QQ sequences were replaced by VL, while single Q was replaced by I. The dimensional stability experiments were expanded by adding the water absorption experiments to more samples, i.e., acetylated RSSP(VLI) and acetylated RSSP(QQQ) fibers, RSP and acetylated RSP fibers, and regenerated *B. mori* silk fibroin and regenerated acetylated *B. mori* silk fibroin fibers. The dimensional stability of the latter four samples was reported previously [22,58]. Finally, in order to clarify what happened in the wet spinning process of RSSP(VLI) from FA solution to fiber formation, we observed three kinds of ^13^C solid-state NMR, i.e., ^13^C refocused insensitive nuclei enhanced by polarization transfer (r-INEPT), ^13^C dipolar decoupled-magic angle spinning (DD/MAS), and ^13^C cross polarization-magic angle spinning (CP/MAS) NMR experiments are performed in the dry and hydrated states [19,22,23,58,59,60,61,62,63,64]. This study, including ^13^C solid-state NMR is the first trial to put spider silk to practical use, providing information about the molecular design of new recombinant spider silk materials with high dimensional stability in water, allowing the use of recombinant spider silk proteins for next-generation biomaterials and materials for textiles.

## 2. Results and Discussion 

### 2.1. ^13^C Solution NMR Spectra of RSSP(VLI) Samples Prepared by Changing the Storage Times in FA

FA is known as a formylating reagent. We previously reported the conformations of RSP and RSSP(QQQ) samples in FA and the formylation of its amino acid side chains [21,23]. These samples consist of Ser and Tyr residues, which are candidates to form formyl groups, but only the Ser side chain was found to be formylated. Thus, we evaluated the formylation of RSSP(VLI) dissolved in FA by ^13^C solution NMR and confirmed the formylation of the Ser side chain.

First, we assigned the ^13^C solution NMR peaks of most of the amino acid residues of RSSP(VLI). Figure 2 shows the ^13^C solution NMR spectrum of the RSSP(VLI) sample in FA after dissolving it for 15 h, together with the peak assignments. All CH_n_ (n = 1–3) groups for most amino acid residues of RSSP(VLI), i.e., Gly, Ala, Pro, Ser, Tyr, Val, Leu, and Ile, were assigned by using ^13^C HSQC and 1D ^13^C NMR spectra. The ^13^C NMR chemical shifts are listed in Appendix A. The ^13^C HSQC spectra of RSSP(VLI) were measured several times continuously to characterize the formylation of the Ser side chain in real time after the dissolution of RSSP(VLI) in FA.

The ^13^C chemical shifts of Ser CαH and Ser CβH_2_ peaks changed after the dissolution of RSSP(VLI) in FA. As shown in Figure 3, unformylated Ser CαH and Ser CβH_2_ peaks were observed in the first ^13^C HSQC spectrum at 0.7 h after dissolution in FA.

The ^1^H and ^13^C chemical shifts were 4.73 ppm and 57.5 ppm for the CαH and 4.08 ppm and 63.3 ppm and 4.17 ppm and 63.3 ppm for the CβH_2_, respectively. The intensity of these peaks gradually decreased with time and became negligible at 7 h after dissolution. On the other hand, formylated Ser peaks were observed at 4.95 ppm and 54.6 ppm for the CαH and 4.62 ppm and 64.6 ppm and 4.66 ppm and 64.6 ppm for the CβH_2,_ and the intensity of these peaks increased with time. The time-dependent changes in intensity for the CαH and CβH_2_ peaks of unformylated and formylated Ser residues are plotted in Figure 4. 

These measurements show that most of the Ser residues in RSSP (VLI) were formylated within 2 h after dissolution in FA. The formylation rate of RSSP (VLI) is faster than that of RSP and RSSP (QQQ) reported previously [21,23]. This may be due to the fact that the reaction temperature of RSSP(VLI) is higher than that of RSP and RSSP (QQQ), being 313 K for RSSP (VLI) but 293 K for RSP and 298 K for RSP (QQQ). It was necessary to dissolve RSSP (VLI) in FA at a higher temperature because the solubility of RSSP (VLI) was lower than that of RSP and RSSP(QQQ).

### 2.2. Stress–Strain Curves of RSSP(VLI) Fiber after Repeated Immersion in Water and Drying of the Fiber

The calculated tensile strength (MPa) and elongation-at-break (%) calculated from the stress–strain curves of RSSP(VLI) fiber prepared by the wet-spinning method from the FA solutions of RSSP(VLI) powder (storage time in FA was 20 h at 40 °C) are shown in Figure 5a.

Stress–strain curves of RSSP(VLI) fiber after hydration treatment, i.e., repeated immersion of the fiber in water and drying at room temperature, are also shown in Figure 5b. The mechanical properties of RSSP(VLI) fiber were compared before and after hydration treatments. The tensile strength (MPa) and elongation-at-break (%) were 134 ± 10 and 19 ± 4 for RSSP(VLI) fiber, respectively, before hydration treatment. The value of the tensile strength (MPa) was lower than the previous value of RSSP(QQQ) fiber, i.e., 191 ± 4 (storage time in FA, 40 h) and 238 ± 5 (storage time in FA, 4 h) [23]. This is potentially due to the difference in the Mw of the samples, i.e., the Mw of RSSP(QQQ) fiber is 210 kDa, which is remarkably higher than the value of Mw 51.6 kDa for RSSP(VLI) fiber. By hydration treatment, the tensile strength (MPa) of RSSP(VLI) fiber decreased from 134 ± 10 to 97 ± 3, but the elongation-at-break (%) increased remarkably from 19 ± 4 to 62 ± 3. These changes are a general matter due to the hydration effect of the silk fibers. Namely, in general, because of the presence of water molecules, the inter-molecular hydrogen bonding of protein fiber such as silk fiber weakens; the molecules are separated from each other, and the space for the movement of the molecules becomes greater, namely, the fiber becomes more plasticized by water. As a result, the tensile strength of silk protein fiber decreases and the elongation-at-break increases by hydration [22,23,24].

### 2.3. Dimensional Stability Experiments of Eight Kinds of Silk Fiber Samples

At first, we compared the dimensional stabilities of RSSP(QQQ) and RSSP(VLI) fibers, determined by repeated immersion in water and drying, as shown in Figure 6. Because changes in the P ratio largely depend on the condition of the fiber formation, we consider only the S ratio for dimensional stability. Here, the definitions of the P and S ratios are given in the Materials and Methods section. The change is remarkable in the S ratio, i.e., the S ratio of RSSP(VLI) fiber becomes almost half that of the RSSP(QQQ) fiber. This is due to the hydrophobic effect as a result of replacing the QQQ amino acid residues with VLI in the recombinant spider silk.

Next, the dimensional stability experiments were expanded for eight kinds of silk fiber, as shown in the histogram in Figure 7.

We added the S ratio of RSP fibers reported previously [22] for recombinant spider silk fibers other than RSSP(QQQ) and RSSP(VLI) fibers. During the acetylation of the silk fiber, it is known that the fiber becomes more hydrophobic due to the acetylation of Ser and Tyr residues in the silk molecules, and therefore, the acetylated RSSP(QQQ), RSSP(VLI), and RSP fibers were selected for dimensional stability experiments. For this purpose, we prepared acetylated RSSP(QQQ) and RSSP(VLI) fibers, as described in the Materials and Methods section. The regenerated *B. mori* silk fiber and acetylated regenerated *B. mori* silk fiber reported previously [58] were also compared with the corresponding data of the recombinant spider silk fibers. The S ratios of both non-acetylated and acetylated recombinant spider silk fibers decreased in the order of RSP, RSSP(QQQ), and RSSP(VLI). The amino acid composition (%) of RSP (Mw = 47.5 kDa) is G:37, Q:19, A:16, P:15, S:6, and Y:4 and that of RSSP(QQQ) (Mw = 210 kDa) is G:31, Q:17, A:20, P:14, S:9, and Y:7. Because the amino acid composition does not change significantly between two samples, the cause of the decrease in the S ratio of the RSSP(QQQ) sample is likely due to the remarkably larger molecular weight compared with that of the RSP sample. The smallest S ratio of the RSSP(VLI) (Mw = 51.6 kDa and G:31, A:20, P:14, S:10, Y:7, V:6, L:6, and I:6) sample is clearly due to the hydrophobic effect as a result of substituting QQQ with VLI, although the Mw is smaller than that of the RSSP(QQQ) sample. In addition, all of the S ratios of the acetylated fibers were smaller than those of the non-acetylated fibers. This is due to the acetylation of the Ser and Tyr residues mentioned above. For *B. mori* silk fibroin fiber, the decrease in the S ratio was relatively larger than the S ratios of recombinant spider silk fibers. Because both Ser and Tyr residues (amino acid composition (%) of S:12 and Y:5.3) are acetylated in *B. mori* silk fibroin fiber [58], the hydrophobic effect by acetylation seems greater than that of recombinant spider silks.

### 2.4. Water Absorption Experiments of Eight Kinds Using Silk Fiber Samples

Eight kinds of silk fiber samples were used for the water absorption experiment, as shown in Figure 8a. The water absorption is generally a reflection of the hydrophilic character of the fiber samples.

To obtain a more detailed insight into the interaction of silk fiber and water molecules, the ^2^H solution NMR relaxation and exchange measurements of water molecules interacting with silk fibers are very useful, as reported previously by us [59,61]. Namely, (a) bulk water outside the fiber, (b) water molecules trapped weakly on the surface of the fiber, (c) bound water molecules located in the inner surface of the fiber, and (d) bound water molecules located in the inner part of the fiber were distinguishable in the experiments. We would like to focus on these experiments in future work. The water adsorption is far greater for RSP fiber, including acetylated RSP fiber, compared with that of other fibers. Figure 8b shows the plot of the S ratio against the value of water absorption. It can be noticed that there is a correlation between the water absorption and S ratio, with only regenerated *B. mori* silk fibroin fiber deviating from the trend of the whole plot. Thus, both the RSSP(VLI) fiber and acetylated RSSP(VLI) fiber show the low water absorption and high dimensional stability of recombinant spider silk fiber.

### 2.5. ^13^C Solid-State NMR Spectra of RSSP(VLI) Powders and Fibers in the Dry and Hydrated States

A variety of techniques, such as X-ray diffraction, Fourier transform infrared/Raman spectroscopy, transmission electron microscopy, and so on, were used to elucidate the structure of spider dragline silk, from its secondary structure, to its molecular arrangement, to its hierarchical structure [1,2,3,4,5,6]. However, the most detailed picture of the structure and dynamics of spider silk at the molecular level was obtained from NMR spectroscopy. The conformation-dependent ^13^C chemical shifts coupled with selective ^13^C labeling can be used effectively to determine the secondary structure of spider silks in an amino-acid-specific manner. Moreover, many kinds of advanced solid-state NMR techniques have been used to obtain the local structure and dynamics of spider silk at atomic resolution [19,20,21,22,23,52,53,65,66,67,68,69,70,71,72]. In this paper, three kinds of ^13^C solid-state NMR, i.e., ^13^C r-INEPT, ^13^C DD/MAS, and ^13^C CP/MAS NMR spectroscopy [19,22,23,58,59,60,61,62,63,64], were used to clarify the secondary structure and dynamics of RSSP(VLI) powder and fiber in the dry and hydrated states.

The ^13^C solid-state NMR spectra of RSSP(VLI) powder before MeOH treatment were first observed in the dry and hydrated states. Figure 9 shows (a) ^13^C r-INEPT, (b) ^13^C DD/MAS, and (c) ^13^C CP/MAS NMR spectra in the hydrated state, together with (d) the ^13^C CP/MAS NMR spectrum in the dry state. Four kinds of spectra are considerably changed and the spectra look sharper gradually from Figure 9(d) to 9(a). The intensity of the ^13^C CP/MAS NMR signal is sensitive to the components of very slow motion (< 10^4^ Hz). In the ^13^C CP/MAS NMR spectrum of RSSP(VLI) in the dry state, the main ^13^C Ala Cβ and Ala Cα peaks were observed at 20.2 (f) and 48.9 ppm (o), respectively (see Table 1). This means that the conformation of the poly-Ala region has an AP-β structure [73,74]. The ^13^C chemical shifts of the Ser Cα and Ser Cβ peaks were 55.0 (r) and 63.2 ppm (v), respectively, indicating that the Ser residue also has an AP-β structure [73,74]. Other peaks from the Gly-rich region were also observed. This means that there are amino acid residues forming inter-molecular hydrogen bonds and/or a spatially densely packed region. These amino acid residues are in a restricted state, even if the conformation of the amino acid residue takes the form of random coil.

When the powder is hydrated in water, water molecules cause a significant increase in the motion of the amino acid residues in the powder. In the ^13^C CP/MAS NMR spectrum Figure 9c, a loss in CP signal occurs because a mobile domain cannot be observed [63,64,70]. Thus, spectrum (c) became sharper than spectrum (d), and the β-sheet peaks of Ala Cα, Cβ, and CO carbons became the main peaks of RSSP(VLI) powder. On the other hand, the ^13^C r-INEPT was sensitive to the component of fast motion (> 10^5^ Hz) in hydrated powder. Therefore, only the peaks that were very mobile in solution NMR could be observed in the ^13^C r-INEPT spectrum (a). Only random coil and sharp peaks induced by water absorption were observed for the Cα and Cβ carbons of Ala and Ser residues. All carbons of other residues in the Gly-rich region were also observed as sharp peaks. Thus, mobile components of RSSP(VLI) powder were observed in water. All of the ^1^H attached carbons in the amino acid residues were observed, although the carbonyl and aromatic peaks with no protons were not observed in the ^13^C r-INEPT spectrum. In the ^13^C DD/MAS spectrum (b), both mobile and immobile components in the hydrated state can be observed, namely, the β-sheet peaks of the Cα and Cβ carbons of Ala and Ser residues were observed together with the mobile peaks observed in the ^13^C r-INEPT spectrum. The carbonyl peaks were also a mixture of both sharp and broad peaks. These results indicate that both rigid β-sheet and flexible random coil structures can be observed in the hydrated state.

Figure 10 shows the (a) ^13^C r-INEPT, (b) ^13^C DD/MAS, and (c) ^13^C CP/MAS NMR spectra of RSSP(VLI) powder after MeOH treatment in the hydrated state, together with (d) the ^13^C CP/MAS NMR spectrum in the dry state. The β-sheet peaks of Ala Cα, Ala Cβ, Ser Cα, and Ser Cβ carbons were observed together with other peaks from the Gly-rich region in spectrum (d). With hydration, the β-sheet peaks of Ala residues were also relatively larger in spectrum (c) compared with spectrum (d). The ^13^C DD/MAS NMR spectrum of RSSP(VLI) powder in water became slightly sharper compared with spectrum (c), but the change was as that shown in Figure 9. The ^13^C r-INEPT spectrum (a) in Figure 10 shows only a small Ala Cβ peak, and other peaks disappeared completely, which is remarkably different from spectrum (a) in Figure 9. Thus, by MeOH treatment, the mobility of RSSP(VLI) powder does not increase significantly, even in the hydrated state. Figure 11 shows the (a) ^13^C r-INEPT, (b) ^13^C DD/MAS, and (c) ^13^C CP/MAS NMR spectra of RSSP(VLI) fiber in the hydrated state, together with (d) the ^13^C CP/MAS NMR spectrum in the dry state. The spectra are very similar between Figure 10 and Figure 11 because both samples were obtained after MeOH treatment.

**Table 1 molecules-27-08479-t001:** ^13^C solid-state NMR chemical shifts (ppm) of RSSP(VLI) powder and fiber.

	Cα	Cβ	Cγ	Cδ
Gly	42.0, 42.8			
	m m			
Ala	50.0 (r.c.)	16.6 (r.c)		
	p	c		
	48.9 (α)	20.2 (β)		
	o	f		
Pro	61.4	31.3	27.0	47.1
	u	k	i	n
Ser	53.3 (r.c.), 52.4 (a)	61.4 (r.c)		
	q q*	u		
	55.0 (α)	63.2 (β)		
	r	v		
Tyr	56.1	36.4		
	s	l		
Val	61.4	29.3	17.8, 18.5	
	u	j	d e	
Leu	53.3	42.8	27.0	22, 24.6
	q	m	i	g h
Ile	59.2	36.4	13.9, 24.6	11.4
	t	l	b h	a

(a). The 52.4 ^(a)^ ppm peak (q*) was assigned to Cα peak of formylated Ser residue. r.c.: random coil and β: β-sheet.

However, there are two different points in Figure 11 compared with Figure 10. Namely, as mentioned above, the Ser residue of RSSP(VLI) is formylated after FA treatment. Therefore, the Ser Cα peak (q*) of formylated RSSP(VLI) was observed at approximately 1 ppm higher than the Ser Cα peak (q) of unformylated RSSP(VLI) (see Table 1). In addition, the formyl peak was newly observed at 161 ppm in Figure 11.

## 3. Materials and Methods

### 3.1. Preparation of RSSP(VLI) Powder Sample

The RSSP(VLI) sample (MW = 51.6 kDa) based on RSSP(QQQ), where all QQ sequences are replaced by VL, while single Q is replaced by I, is produced using *E. coli*. and purified with a Ni column [17,18]. The His-tag is attached to the N-terminal side of the amino acid sequence and used for sample purification. The primary structure is shown in Figure 1. We tried to produce RSSP(VLI) with Mw = 210 kDa, because RSSP(QQQ) with Mw = 210 kDa was used for a comparison in further experiments, but we could not obtain sufficient amounts of the samples. The RSSP(VLI) powder was used for the observation of solution NMR, and the powder samples before and after MeOH treatments were used for ^13^C solid-state NMR observations. All of the reagents used here were purchased from FUJIFILM Wako Pure Chemical Corporation, Japan.

### 3.2. Preparation of Acetylated RSSP(VLI) and Acetylated RSSP(QQQ) Powder Samples

For acetylation, RSSP(VLI) powder (1.0 g) was dissolved in dimethylformamide (DMF, 40 mL) with LiCl (1.6 g) at 80 °C with a stirrer [22]. Then, acetic anhydrate (5 mL) was added and stirred at 80 °C for 6 h. The DMF solution was poured in MeOH (400 mL). The precipitate was collected by centrifugal separation and washed a few times with MeOH. The precipitate suspension in distilled water was dried by lyophilization and then dried at 80 °C overnight. The acetylation of RSSP(QQQ) powders was performed with a similar method. These two acetylated samples, acetylated RSSP(VLI) and RSSP(QQQ), were used for dimensional stability and water absorption experiments after becoming fibrous together with the non-acetylated RSSP(VLI) and RSSP(QQQ) fibers.

### 3.3. ^13^C Solution NMR Observation of RSSP(VLI) Sample

The RSSP(VLI) powder was dissolved in formic acid-d_1_ (Cambridge Isotope Laboratories, Inc., Andover, MA.) to a concentration of 5 *w*/*v*% and stored in a 5 mm Shigemi microtube. The solution NMR experiments were performed on a JEOL Resonance ECZ500 spectrometer equipped with a HCN triple resonance inverse probe at 313 K. The assignments of the ^13^C peaks were accomplished using ^1^H-^13^C HSQC and 1D ^13^C NMR spectra. In order to characterize the formylation of RSSP(VLI) in FA, the ^1^H-^13^C HSQC spectra were observed as a function of time after the RSSP(VLI) powder was dissolved in FA. Spectra were processed using JEOL Delta and NMR Pipe. The TMS proton signal at 0 ppm was used as a chemical shift reference for ^1^H signals. The ^13^C chemical shifts were indirectly referenced by using the ^1^H chemical shift value of TMS.

### 3.4. Preparation of RSSP(VLI) Fiber, Acetylated RSSP(VLI), and Acetylated RSSP(QQQ) Fibers 

The preparation of RSSP(VLI) fiber was performed as follows. The RSSP(VLI) powder was dissolved in FA (conc. 20 *w*/*v*%) with a stirrer for 16 h at 40 °C and dissolution using rotation/revolution mixer (Awatori Rentaro^®^ ARE-310: THINKY Co., Ltd., Tokyo, Japan) within 4 h. After the FA solution of RSSP(VLI) was filtered with a 5 m steel filter, the FA solution was extruded through a stainless steel spinneret with 0.2 mm inner diameter into the MeOH coagulation bath at room temperature without an air gap [22,23,58,75,76,77]. The coagulation time of the fiber in MeOH was set for 1 min, and then, the fiber was stretched by 4 times with a manual uniaxial stretching machine (Imoto machinery Co. Ltd., Japan) and dried at room temperature. The fibers were used for further experiments, i.e., the determination of the mechanical properties, dimensional stability evaluation, water absorption experiments, and the ^13^C solid-state NMR measurements. The fibers of acetylated RSSP(VLI) and acetylated RSSP(QQQ) powders were prepared according to a similar method, mentioned above.

### 3.5. Mechanical Property Measurements of RSSP(VLI) Fibers

The stress–strain curves of the fibers were measured with an EZ-Graph tensile testing machine (EZ-Graph, SHIMADZU Co. Ltd. Japan) at room temperature with a 5 Newton (N) load cell. In the measurement of the stress–strain curve, both ends of the fibers were mounted on Scotch tape with a base length of 20 mm and fixed with ethyl cyanoacrylate [22,23,58,65,66,67]. The rate of crosshead was 3 mm/min on samples of 20 mm length. The breaking strength (MPa) measured as the highest stress value attained during the test was calculated by dividing the cross-sectional area of the fiber. The elongation-at-break (%) was measured as the change in length divided by the initial length. Each value was obtained by averaging over 3–6 measurements. Before starting the tensile test, the diameters of these fibers were measured with an optical microscope (KEYENCE BIOREVO BZ-9000, Japan). The stress–strain curves of the fibers were also observed after the dimensional stability measurements described below. The ends of the fibers were fixed to maintain the length constant at room temperature before the stress–strain experiments.

### 3.6. Dimensional Stability Experiments with Eight Kinds of Silk Fiber Samples by Repeated Immersion in Water and Drying

Eight kinds of silk fibers, i.e., the RSSP(VLI) (Mw:51.6 kDa) and acetylated RSSP(VLI) fibers, RSSP(QQQ) (Mw:210 kDa) and acetylated RSSP(QQQ) fibers, and RSP (Mw:47.5 kDa) and acetylated RSP fibers reported previously [22], and regenerated *B. mori* silk fibroin and regenerated acetylated *B. mori* silk fibroin fibers, also reported previously [58], were used for the dimensional stability measurements. The dimensional stabilities were determined by repeated immersion in water and drying at room temperature as follows [22,58]. A bundle of 24 fibers was loaded onto a 7.84 mN weight installed at the bottom edge of the bundle. The length of the bundle in the dry state, L_dry0_, was determined. After immersing the bundles in water for 30 min, the length in the hydrated state, L_hydrated1_, was determined. Here, the primary contraction, or P ratio, of the bundles is defined as the change in length when the bundles are first immersed in water, expressed by the following formula: (L_hydrated1_ − L_dry0_) /L_dry0_ [22,58]. Then, the bundles were dried at room temperature for 30 min and L_dry2_ is determined. This hydration and drying treatment was repeated three times. The averaged value over (L_hydrated3_ − L_dry2_) /L_dry2_, (L_hydrated3_ − L_dry4_)/L_dry4_, (L_hydrated5_ − L_dry4_) /L_dry4_ and (L_hydrated5_ − L_dry6_)/L_dry6_ was defined to determine the secondary contraction, S ratio [22,58]. We compared only the S ratio as dimensional stability among samples because the P ratio changed greatly depending on the condition of the fiber formation.

### 3.7. Water Absorption Experiments of Eight Kinds Using Silk Fiber Samples

Eight kinds of silk fibers were used for water absorption experiments. Approximately 35 mg of each dry sample of the eight kinds of silk fiber was weighed after drying the samples at 80 °C for 16 h. Then, these samples were pressed at 20 MPa for 5 min using a tablet modeling machine for infrared measurement, and a tablet of diameter ca. 10 mm and thickness ca. 1mm was obtained. The weight, A mg, of the dry tablet was measured. Then, the tablet was immersed in water for 30 min and the water quickly wiped from the surface of the tablet. The weight, B mg, of the wet tablet was measured. The water absorption (%) of the tablet was calculated using the equation [(B – A)/A] × 100 (%).

### 3.8. ^13^C r-INEPT, ^13^C CP/MAS and ^13^C DD/MAS NMR Measurements of RSSP(VLI) Powders and Fibers in the Dry and Hydrated States 

The ^13^C solid-state NMR spectra of RSSP(VLI) powders before and after MeOH treatments, as well as the RSSP(VLI) fibers, were observed in the dry and hydrated states using a Bruker Avance 400 NMR spectrometer with a 4 mm double resonance MAS probe and a MAS rate of 8.5 kHz at room temperature. For the NMR observations in the hydrated state, the samples were carefully inserted into a zirconia rotor, sealed with polytetrafluoroethylene (PTFE) insert to prevent the dehydration of the hydrated samples [19,22,23,33,34,35,36,37,38,39,40,41,42,43]. Typical parameters for the ^13^C CP/MAS NMR experiments were a 3.5 μs ^1^H 90° pulse, a 1 ms ramped CP pulse with 71.4 kHz rf field strength, two pulse phase modulation (TPPM) ^1^H decoupling during acquisition, 2176 data points, 8 k scans, 310 ppm sweep width, and a 4 s recycle delay. Lorentzian line broadening of 20 Hz was applied prior to Fourier transformation. To observe both the mobile and immobile parts of PSSP(VIL), the ^13^C DD/MAS NMR method was used. In the ^13^C DD/MAS NMR experiments, ^13^C nuclei were excited using a single 90° pulse, and ^13^C free induction signals were subsequently acquired under a high power ^1^H decoupling pulse. A recycle delay of 5 s was used to prevent the saturation of ^13^C NMR signals. A repetition time of 5 s was selected because the reduction in ^13^C NMR signals by saturation caused by long ^13^C spin lattice relaxation time in the rigid part was negligible, as indicated in our previous repetition time variation experiments [59]. Details of the NMR conditions for the ^13^C DD/MAS NMR experiments were described previously [59], including 8 k scans, a recycle delay of 5 s, and a ^13^C 90° pulse of 3.5 μs. Typical experimental parameters for the r-INEPT NMR experiments include 3.5 μs ^1^H and 3.6 μs ^13^C pulses, an inter-pulse delay of 1/4^1^J_CH_ (^1^J_CH_ =145 Hz), a refocusing delay of 1/6^1^J_CH_ or 1/3^1^J_CH_, TPPM ^1^H decoupling during acquisition, 1438 data points, 4 k scans, 200 ppm sweep width, and a 3.5 s recycle delay. The ^13^C chemical shifts were calibrated externally through the methylene peak of adamantane observed at 28.8 ppm relative to TMS at 0 ppm.

## 4. Conclusions

A new recombinant spider silk protein, RSSP(VLI) based on the RSSP(QQQ), where all QQ sequences are replaced by VL, while single Q is replaced by I, is produced with *E. coli* to overcome a critical defect: the low dimensional stability of RSSP(QQQ) fiber in water, preventing its practical use. The fiber was prepared from powder using formic acid as the spinning solvent and methanol as the coagulant solvent during the wet spinning process. A higher dimensional stability of RSSP(VLI) fiber than other silk fibers was obtained. A remarkable decrease in the motion of the fiber in water was evident by ^13^C solid-state NMR studies. This study, including ^13^C solid-state NMR is the first trial to put spider silk to practical use, providing information for the molecular design of new recombinant spider silk materials with high dimensional stability in water, allowing the use of recombinant spider silk proteins in next-generation biomaterials and materials for textiles.

## Figures and Tables

**Figure 1 molecules-27-08479-f001:**
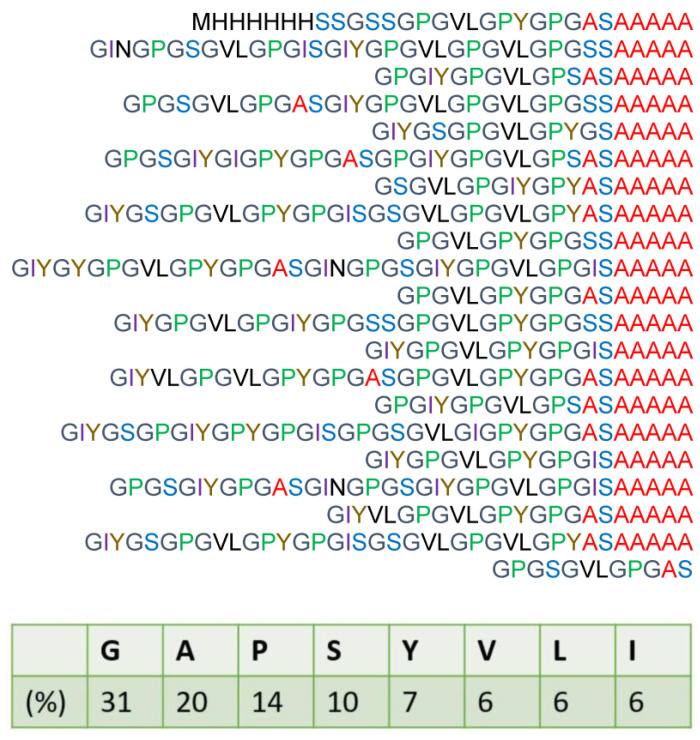
Amino acid composition and primary structure of recombinant spider silk protein RSSP(VLI) based on the primary sequence of ADF-3 *Araneus diadematus* silk protein with the Gln site replaced by Val, Leu, and Ile.

**Figure 2 molecules-27-08479-f002:**
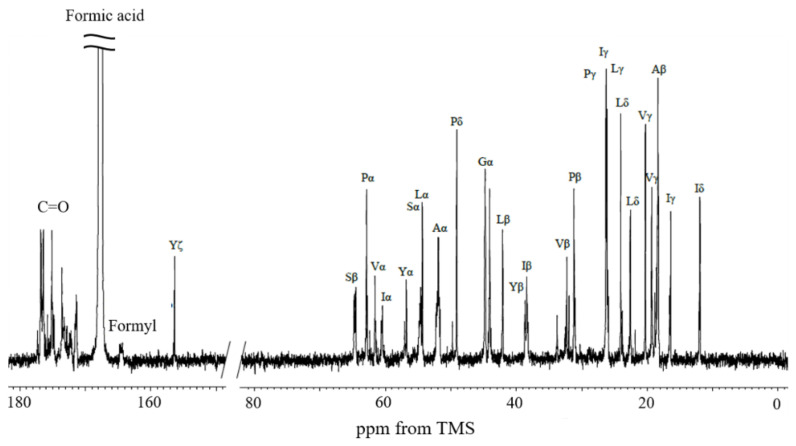
^13^C solution NMR spectrum of RSSP(VLI) sample in formic acid after dissolving it for 15 h together with the peak assignments.

**Figure 3 molecules-27-08479-f003:**
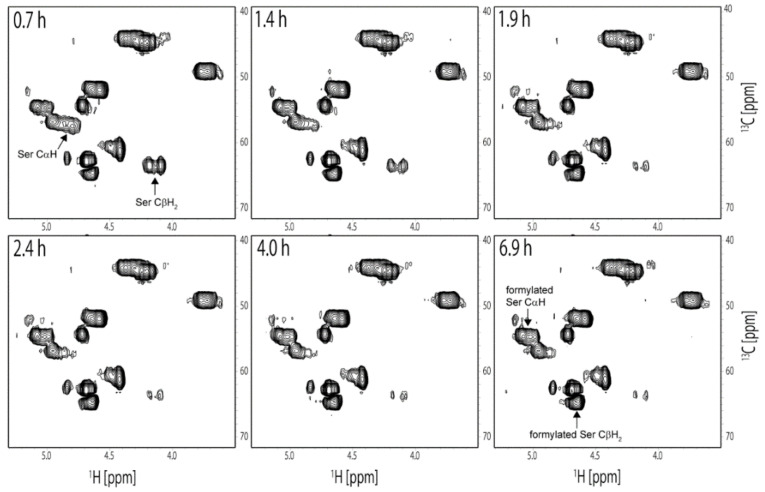
^13^C HSQC spectra of RSSP(VLI) measured several times continuously to observe the formylation in real time after the dissolution of RSSP(VLI) in FA. The elapsed time from start of the measurements is denoted in the top left of each box. Peaks derived from unformylated and formylated Ser residues are denoted in the spectra.

**Figure 4 molecules-27-08479-f004:**
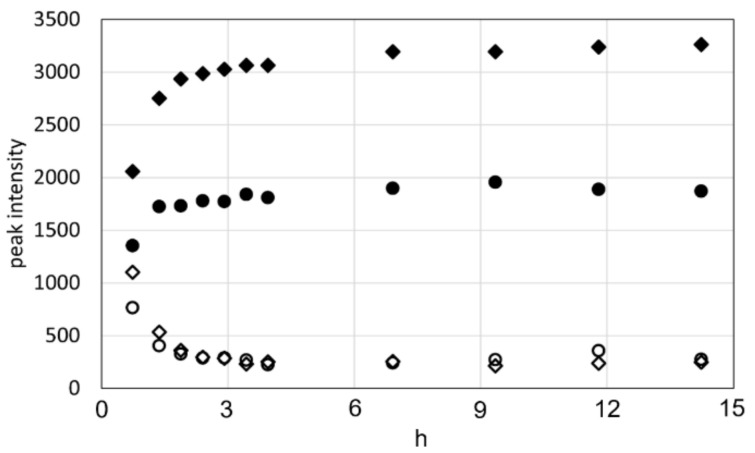
Peak intensity for the CαH and CβH_2_ protons of unformylated and formylated Ser residues with time after the dissolution of RSSP(VLI) in formic acid. (● formylated Hα, ◆ formylated Hβ, 〇 unformylated Hα, ◇ unformylated Hβ).

**Figure 5 molecules-27-08479-f005:**
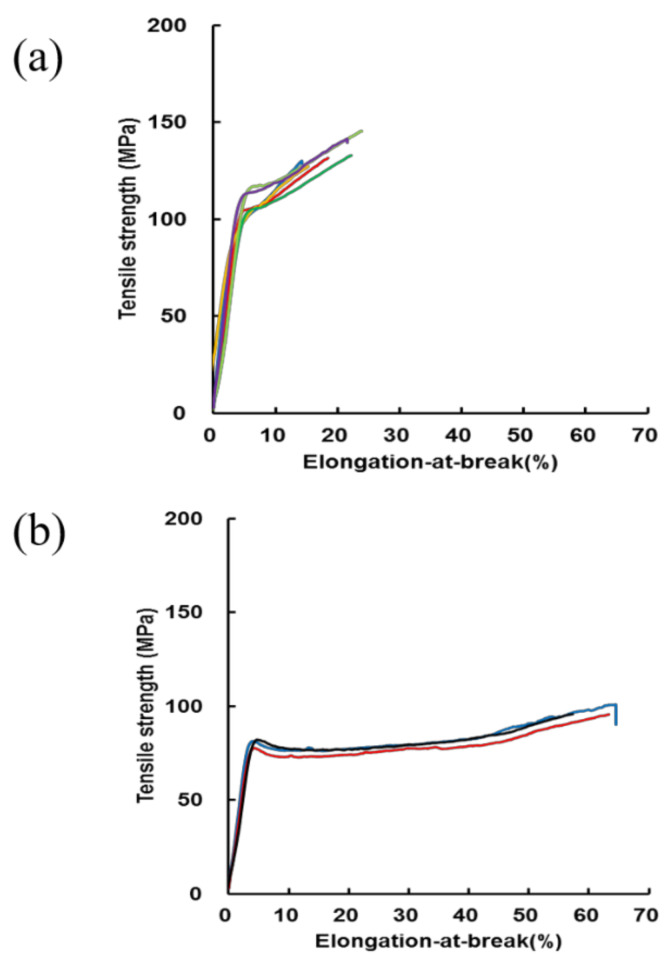
Stress–strain curves of (**a**) RSSP(VLI) fiber prepared by the wet spinning method from FA solutions of RSSP(VLI) powder (storage time in FA was 40 h at 40 °C) before hydration treatment and (**b**) RSSP(VLI) fiber after dimensional stability measurement, i.e., repeated immersion of the fiber in water and drying at room temperature. The calculated tensile strength (MPa), elongation-at-break (%), and diameter (μm) were (**a**) 134 ± 10, 19 ± 4, and 61.9 and (**b**) 97 ± 3, 62 ± 3, and 59.6, respectively.

**Figure 6 molecules-27-08479-f006:**
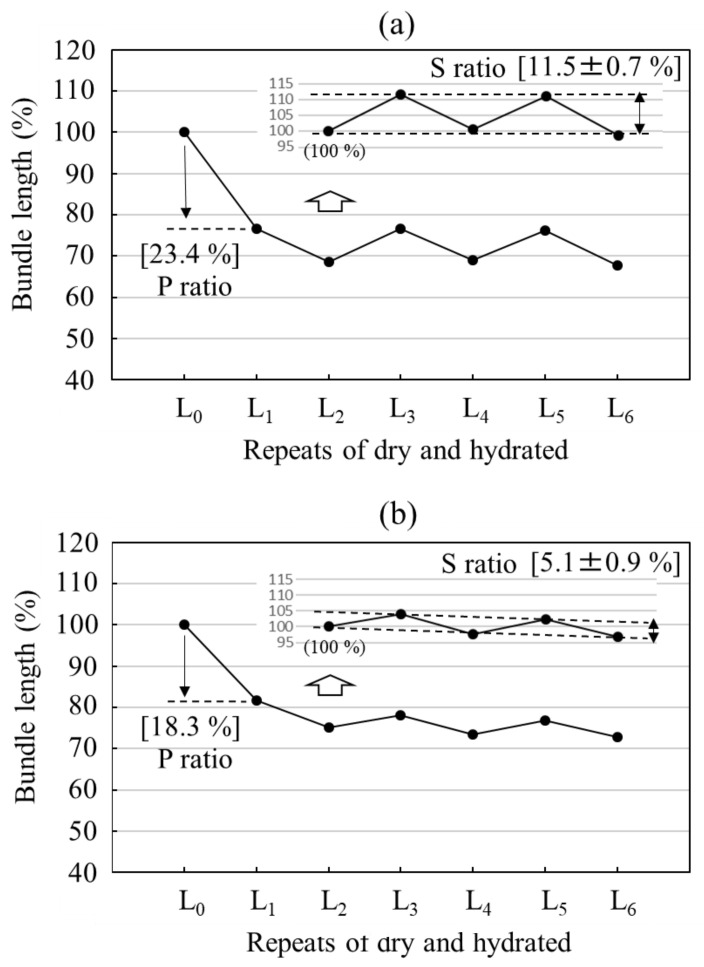
Dimensional stabilities of regenerated fibers of (**a**) RSSP(QQQ) and (**b**) RSSP(VLI) after repeated immersion in water. L0:L_dry0,_ L1:L_hydrated1,_ L2:L_dry2,_ L3:L_hydrated3_, L4:L_dry4_, L5:L_hydrated5_, and L6:L_dry6_.

**Figure 7 molecules-27-08479-f007:**
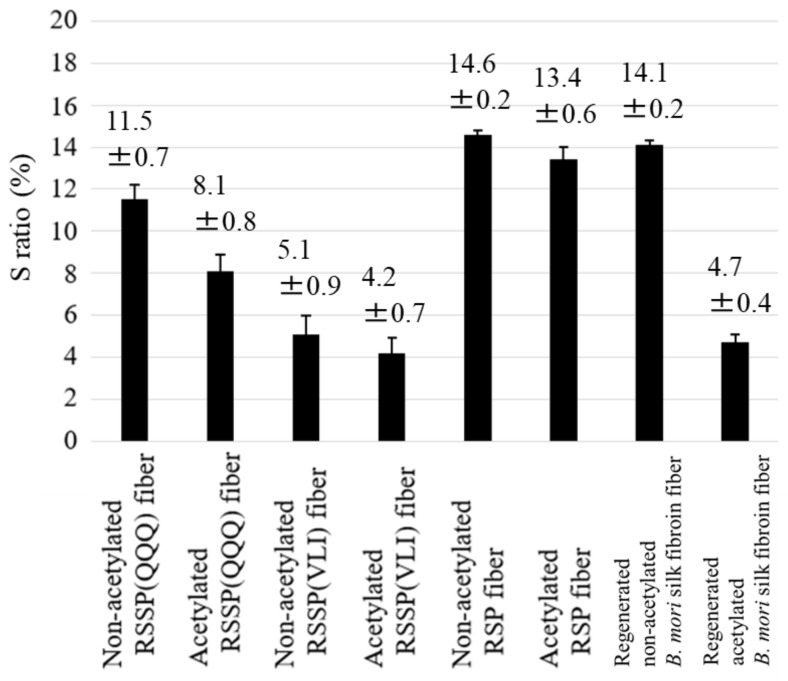
The secondary contraction, S ratio, of eight kinds of silk fiber, i.e., non-acetylated and acetylated fibers of RSSP(QQQ), RSSP(VLI), RSP, and regenerated *B. mori* silk. The S ratio is defined in the text.

**Figure 8 molecules-27-08479-f008:**
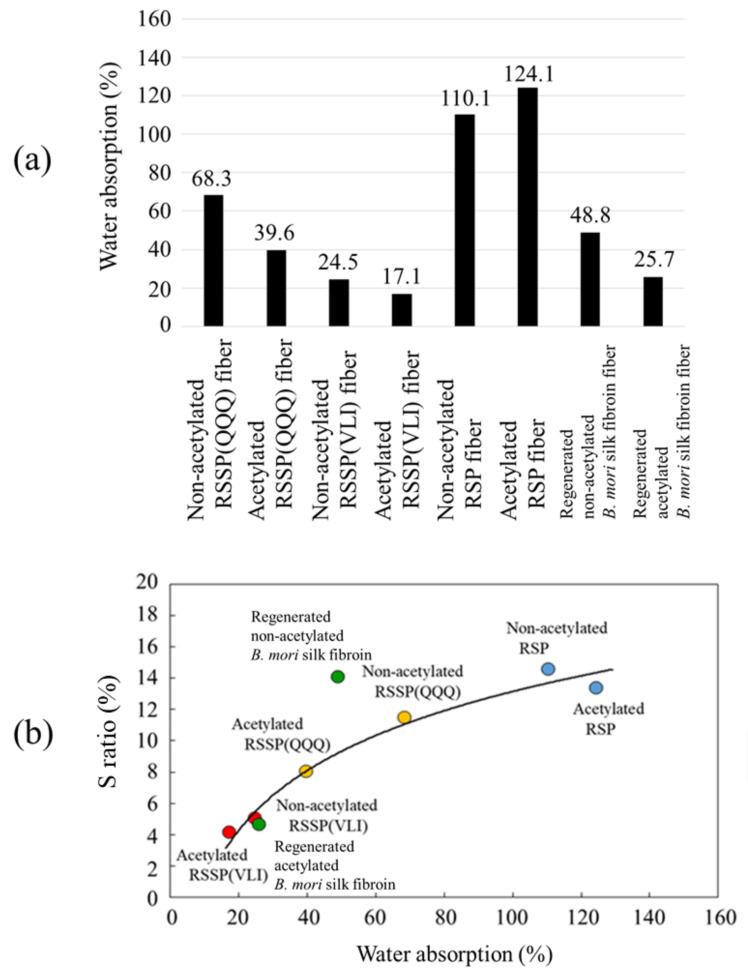
(**a**) The water absorption (%) and (**b**) the correlation between the water absorption (%) and the S ratio (%) observed for eight kinds of silk fibers, i.e., non-acetylated and acetylated fibers of RSSP(QQQ), RSSP(VLI), RSP, and regenerated *B. mori* silk. Water absorption and S ratio are defined in the text. The data of both non-acetylated and acetylated fibers of RSSP(QQQ), RSSP(VLI), RSP, and regenerated *B. mori* silk are the circles colored in orange, red, blue, and green.

**Figure 9 molecules-27-08479-f009:**
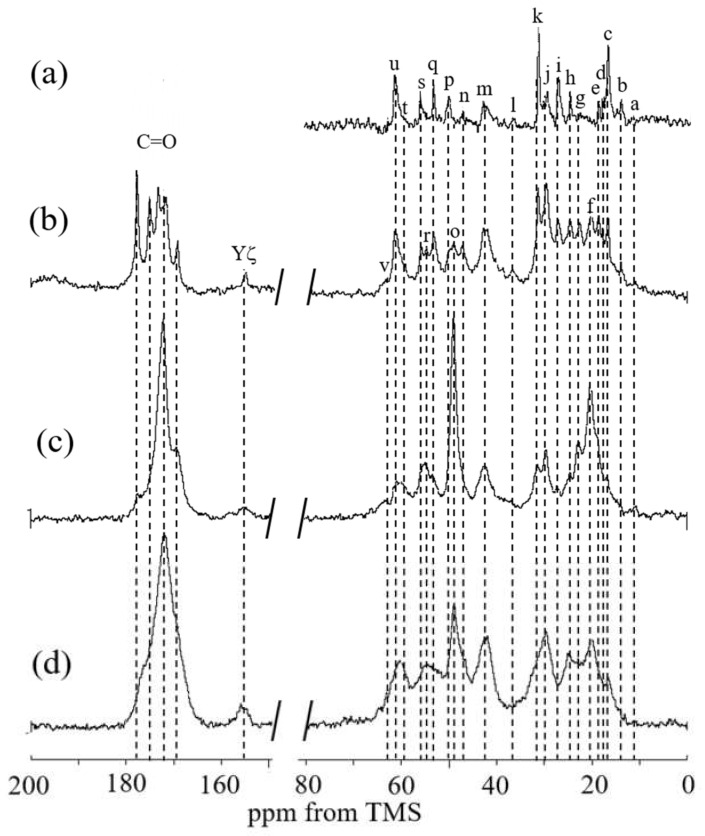
(**a**) ^13^C r-INEPT, (**b**) ^13^C DD/MAS, and (**c**) ^13^C CP/MAS NMR spectra of RSSP(VLI) powders before methanol treatment in the hydrated states and the (**d**) ^13^C CP/MAS NMR spectrum in the dry state. Peak assignments are as follows: a: Iδ, b: Iγ, c: Aβ(r.c.), d: Vγ, e: Vγ, f: Aβ(β), g: Lδ, h: Iγ, Lδ, i: Pγ, Lγ, j: Vβ, k: Pβ, l: Iβ, Yβ, m: Gα, Lβ, n: Pδ, o: Aα(β), p: Aα(r.c.), q: Lα, Sα(r.c.), r: Sβ(β), s: Yα, t: Iα, u: Pα, Sβ(r.c.), Vα, and v: Sα(β).

**Figure 10 molecules-27-08479-f010:**
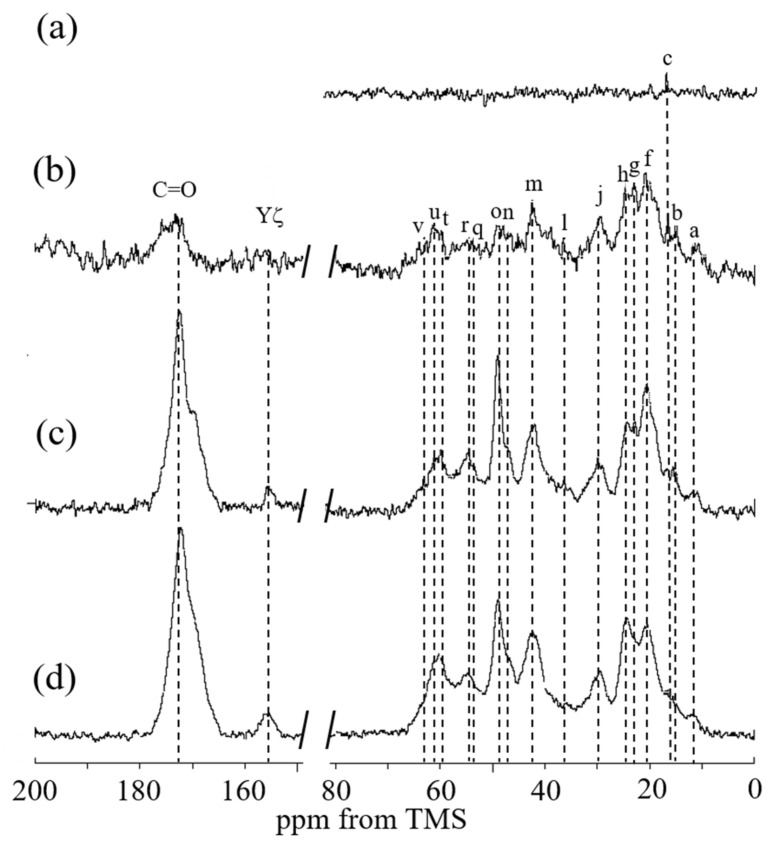
(**a**) ^13^C r-INEPT, (**b**) ^13^C DD/MAS, (**c**) ^13^C CP/MAS NMR spectra of RSSP(VLI) powder after methanol treatment in the hydrated states, and (**d**) ^13^C CP/MAS NMR spectrum in the dry state. The peak assignment is shown in Figure 9.

**Figure 11 molecules-27-08479-f011:**
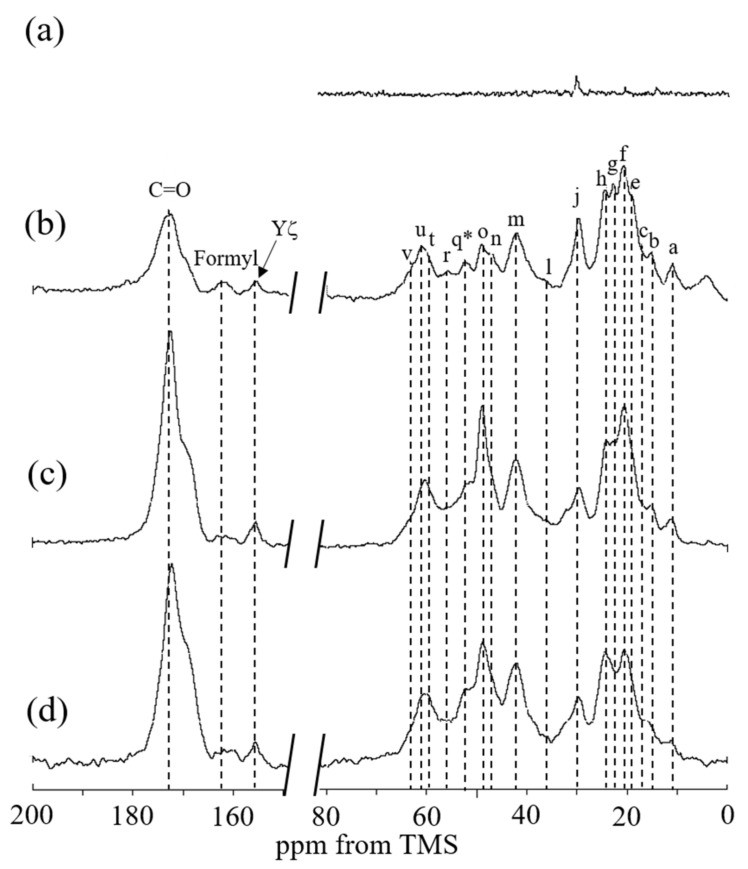
(**a**) ^13^C r-INEPT, (**b**) ^13^C DD/MAS, and (**c**) ^13^C CP/MAS NMR spectra of RSSP(VLI) fiber in the hydrated states and (**d**) ^13^C CP/MAS NMR spectrum in the dry state. The peak assignment is shown in Figure 9. A peak at around 30 ppm in the spectrum (**a**) was assigned to impurity.

## Data Availability

The data presented in this study are available in this article.

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
