# Peer review of "Recombinant Spider Silk Fiber with High Dimensional Stability in Water and Its NMR Characterization"

_molecules, 2022, doi:10.3390/molecules27238479_

Round 1

Reviewer 1 Report

1. The authors should improve the introduction; in my opinion, the main purpose of the experiment is not sufficiently clearly defined. Where can that kind of fiber be used? What is innovative in this research? 

2. In the conclusion, the authors summarize the research, but similarly to the introduction, they neither indicate the use nor the uniqueness/novelty of the introduced solutions to obtain the represented fibers. There are also no references to similar studies.

Author Response

Response to reviewer #1

  1. The authors should improve the introduction; in my opinion, the main purpose of the experiment is not sufficiently clearly defined. Where can that kind of fiber be used? What is innovative in this research? 

Thank you for useful comments which are very helpful to clarify main purpose of our paper.

The authors should improve the introduction; in my opinion, the main purpose of the experiment is not sufficiently clearly defined.

The main purpose of the experiment is clearly defined and we added the statements at lines 62-66 as follows.

In this paper, new recombinant spider silk protein, RSSP(VLI) (Figure 1) based on the RSSP(QQQ) [23] where all QQ sequences are replaced by VL, while single Q is replaced by I is produced with E. coli to overcome critical defect :  low dimensional stability of RSSP(QQQ) fiber in water. In addition, to clarify the structural and dynamical change in the fiber formation process of the recombinant silk protein at molecular level, solid-state NMR was used.

Where can that kind of fiber be used?

This kind of fiber can be used as biomaterials and materials for textile because of their outstanding mechanical properties and biocompatibility of spider dragline silk. However, biomaterials are generally used in water and therefore, it is required to maintain dimensional stabilities of biomaterials in water. In addition, textile is generally required to overcome some inferior textile performance of the fiber, i.e., low water repellency, vulnerable to friction, low dimensional stability, prone to wrinkles and so on, and therefore dimensional stabilities in water are also necessary. Therefore, the recombinant spider silk fiber, RSSP(VLI) with high dimensional stability in water can be used as these materials.

What is innovative in this research? 

This series of research, i.e., change the amino acid sequence from hydrophilic Q to hydrophobic VLI to overcome critical defect : low dimensional stability of spider silk in water, fiber formation by wet spinning method from the powder dissolved in formic acid and obtaining the structural and dynamical information in molecular level occurred in wet spinning process using solid-state NMR are the first trial to put spider silk to practical use and no such studies have been reported as far as we know.

  1. In the conclusion, the authors summarize the research, but similarly to the introduction, they neither indicate the use nor the uniqueness/novelty of the introduced solutions to obtain the represented fibers. There are also no references to similar studies.

We revised Conclusion (lines 588-599) as follows.

New recombinant spider silk protein, RSSP(VLI) based on the RSSP(QQQ) where all QQ sequences are replaced by VL, while single Q is replaced by I is produced with E. coli to overcome critical defect : low dimensional stability of RSSP(QQQ) fiber in water and to use the spider silk practically. The fiber was prepared from the powder using formic acid as spinning solvent and methanol as coagulant solvent in the wet spinning process. The higher dimensional stability of RSSP(VLI) fiber than other silk fibers was obtained. Remarkable decrease in motion of the fiber in water becomes clear by 13C solid-state NMR studies. This series of research including 13C solid-state NMR is the first trial to put spider silk to practical use and are provided information about molecular design of new recombinant spider silk materials with high dimensional stability in water to use the recombinant spider silk proteins for next-generation biomaterials and materials for textile.

Author Response

Response to reviewer #2

  1. The morphologies of the recombinant spider silk fiber and no recombinant one should be given and described, such as, photo pictures or SEM ones.
  2. In this manuscript, the recombinant spider silk fiber is also regenerated fiber given by wet spinning. The natural fibers were generated and showed different properties, for example, cotton and Lyocell. It will be better to give the comparison of fibers’ properties before and after regenerated in the manuscript.

I am sorry we already finished this work more than 1 year ago. Therefore, we hane no sample this time and we cannot observe the photo pictures or SEM ones. In addition, we only obtained the powder of recombinant spider silk produced by E. coli and then prepared the fiber with wet spinning method. Therefore, we did not obtain the natural fiber for comparison of fibers’ properties with natural fiber’ones.

  1. There are many typographical errors, some of which may be caused by converting to pdf format. -The sentences were divided by all figures. –

“Figure 2 13C solution NMR spectrum of…” in line 299, page 7, may be “Figure 2. Solution NMR 13C spectrum of…”. - The font and size in all Figures should fit for the main body, and could not be distortion or deformation. –

Line 330-331, page 10, more “Figure 6” –

Line 337-338, page 11, font or size is mismatching - The content of legends is too much and some explanation moved to the main text. –

Line 463, page 16, more “Figure 10…..” –

The mismatching occurred when the publishers changed our PDF to the style of Molecules. Therefore, we revised our draft to the style of Molecules by ourselves in the revised paper. The content of legend of Figure 6 was shortened.

Line 537, Page20, “https:doi:org/10.1038/s41467-022-31883-3.” Doi mode different from other ref. The whole manuscript should be revised carefully. 4. The reference is also too many, which is suggested to be refined and simplified.

For example, in Line 435, page 15, the citing description is “so AP-b structure [89-94].”, and there 1 or 2 ref may be enough

According to reviewer’s suggestion, the number of references was reduced significantly. Namely, 17 references were deleted in the revised paper. The references of [89-94] were also reduced to two references.

Reviewer 3 Report

The authors present an interesting study on the dimesional stability of spider silk fiber in water.

The manuscript is probably publishable after minor revisions on the solid state NMR part.

In particular, the authors should revise the following points:

Lines 203 and 204. The length of 13C 90 pulse is 3.5 microseconds and not 3.5 milliseconds. Also the lengths of the pulses 1H and 13C used in r-INEPT are most probably 3.5 microseconds and 3.6 microseconds, and not 3.5 milliseconds and 3.6 milliseconds.

Line 201: it should be clearly stated that 13C DD/MAS sequence is a single pulse excitation sequence.

Line 202: is the recycle delay of 5 s sufficient to allow a full recover of the 13C magnetization in 13C DD/MAS experiments? In rigid domains, the relaxation time of 13C may be quite long. Did the authors record spectra applying longer recycle delays?

Lines 442-444. The loss of signal compared to the CP of the dry sample is due to the fact that mobile domains cannot be observed. I would not use the word “consequently” because the  mobile domains are the cause for the loss of signal.

Author Response

Response to reviewer #3

Lines 203 and 204. The length of 13C 90 pulse is 3.5 microseconds and not 3.5 miliseconds. Also the lengths of the pulses 1H and 13C used in r-INEPT are most probably 3.5 microseconds and 3.6 microseconds, and not 3.5 milliseconds and 3.6 millisecond.

We have changed 3.5, 3.5 and 3.6 msec to msec at lines 228 and 229.

Line 201: It should be clearly stated that 13C DD/MAS sequence is a single pulse excitation sequence.

We added the statements at line 220 as follows

To observe both mobile and immobile parts of PSSP(VIL) powder, 13C DD/MAS NMR method was used. In the 13C DD/MAS NMR experiments, 13C nuclei are excited by using single 90º pulse and subsequently 13C free induction signals were acquired under high power proton decoupling pulse.

Line 202: is the recycle delay of 5 s sufficient to allow a full recover of the 13C magnetization in 13C DD/MAS experiments? In rigid domains, the relaxation time of 13C may be quite long. Did the authors record spectra applying longer recycle delays?

We added the statements at line 223 as follows

Recycle delay of 5 sec was used to prevent the saturation of 13C NMR signals. 5 sec of repetition time was selected because the reduction of 13C NMR signals by saturation caused by long 13C spin lattice relaxation time in the rigid part is negligible as indicated in our previous repetition time variation experiments [59].

Line 442-444. The loss of signal compared to the CP of the dry sample is due to the fact that mobile domains cannot be observed. I would not use the word “consequently” because the mobile domains are the cause for the loss of signal.

Line 473: “a loss in CP signal occurs and consequently a mobile domain cannot be observed “ was changed to “a loss in CP signal occurs because a mobile domain cannot be observed.”

Round 2

Reviewer 2 Report

The references should be revised, especially doi formats. 
For example:

6. https:doi:org/10.1038/s41467-022-31883-3

11. doi:....

12. https://doi....

Author Response

we have corrected it.